

# A technique for the measurement of organic aerosol hygroscopicity, oxidation level, and volatility distributions

Kerrigan P. Cain[1] and Spyros N. Pandis[1,2,3]

[1]Department of Chemical Engineering, Carnegie Mellon University, Pittsburgh, USA

[2]Institute of Chemical Engineering Sciences, ICE-HT, Patras, Greece

[3]Department of Chemical Engineering, University of Patras, Patras, Greece

*Correspondence to*: Spyros N. Pandis (spyros@andrew.cmu.com)

**Abstract.** Hygroscopicity, oxidation level, and volatility of organic pollutants are three crucial
properties that determine their fate in the atmosphere. This study assesses the feasibility of a
novel measurement and analysis technique to determine these properties of organic aerosol
components at the same time and to establish their relationship. The proposed experimental
setup utilizes a cloud condensation nuclei counter to quantify hygroscopic activity, an aerosol
mass spectrometer to measure the oxidation level, and a thermodenuder to evaluate the
15 volatility. The setup was first tested with secondary organic aerosol (SOA) formed from the
ozonolysis of α-pinene. The results of the first experiments indicated that, for this system, the
less volatile SOA contained species that had on average lower O:C ratios and hygroscopicities.
In this SOA system, both low and high volatility components can have comparable oxidation
levels and hygroscopicities. The method developed here can be used to provide valuable
insights about the relationships among organic aerosol hygroscopicity, oxidation level, and
volatility.

## 1 Introduction

Anthropogenic activities, such as fuel combustion, as well as biogenic sources, such as
emissions from vegetation, can introduce particles and particle precursors into the atmosphere.
These airborne particles have been identified as a factor contributing to cardiovascular and
respiratory diseases (Pope, 2000; van Eeden et al., 2005) and an increased risk for acute
morbidity and mortality (Dockery, 2001; Miller et al., 2007; Brook et al., 2010). In addition,
atmospheric aerosols influence the Earth's radiation balance directly by scattering and
absorbing solar radiation and indirectly by serving as cloud condensation nuclei (CCN).




Despite their major role in the Earth's energy balance, their net effect on climate is one of the major uncertainties in the climate change problem (IPCC, 2013).

In most areas, organic compounds represent anywhere from 20–90 % of the submicron aerosol mass (Murphy et al., 2006; Zhang et al., 2007). Organic aerosol (OA) is traditionally

classified either as primary (POA) or secondary (SOA). POA refers to organic compounds that are emitted to the atmosphere directly in the particulate phase. SOA refers to particulate matter produced by gas-to-particle conversion processes. In general, intermediate volatility, semi-volatile, and volatile organic compounds undergo oxidation in the atmosphere and form products that can condense on pre-existing particles.

Three of the most important properties of OA are hygroscopicity, oxidation level, and volatility. Hygroscopicity, often expressed by the hygroscopicity parameter, $\kappa$ (Petters and Kreidenweis, 2007), is a measure of a particle's ability to absorb water and form cloud droplets. Oxidation level often provides an indication of the age of the OA in the atmosphere. It is expressed as the oxygen to carbon (O:C) ratio or, more accurately, the average carbon oxidation

state ($OS_C$). Volatility determines the partitioning of organic compounds between the gas and particle phases. The one-dimensional volatility basis set (1D-VBS, Donahue et al., 2006) has been proposed as a framework for the description of the evolution of the OA volatility distribution using logarithmically spaced bins.

Several studies have shown that SOA from the ozonolysis of $\alpha$-pinene is reasonable

CCN material using supersaturated conditions (VanReken et al., 2005; Huff Hartz et al., 2005; Prenni et al., 2007; King et al., 2007; Engelhart et al., 2008; King et al., 2009; Wex et al., 2009; Massoli et al., 2010; Kuwata et al., 2011; Frosch et al., 2011). These studies all reported similar $\kappa$ values based on CCN measurements, ranging from 0.07–0.15. Huff-Hartz et al. (2005) found that SOA produced from monoterpenes was more hygroscopic than SOA from sesquiterpenes.

Both King et al. (2007) and Kuwata et al. (2011) observed that the CCN behavior of this SOA system was dependent on the OA mass concentration. Frosch et al. (2011) found that $\kappa$ for $\alpha$-pinene ozonolysis SOA increases with chemical aging.

Hygroscopic properties of $\alpha$-pinene ozonolysis SOA have also been studied using subsaturated conditions (Prenni et al., 2007; Wex et al., 2009; Poulain et al., 2010; Massoli et

al., 2010; Tritscher et al., 2011). These studies determined a hygroscopic growth factor (HGF) to estimate $\kappa$, reporting values from 0.01–0.08. Massoli et al. (2010) used both a hygroscopic tandem differential mobility analyzer (H-TDMA) and a CCN counter while studying SOA from the ozonolysis of $\alpha$-pinene and found that the $\kappa$ at subsaturated conditions (estimated from HGF measurements) was 20–50 % lower than that based on CCN measurements at



supersaturated conditions. A number of explanations have been proposed for this behaviour including increasing dissolution of SOA components at supersaturated conditions, surface tension effects, etc. (Massoli et al., 2010).

The oxidation level of α-pinene ozonolysis SOA has mainly been studied through the use of high resolution mass spectrometry (Bahreini et al., 2005; Alfarra et al., 2006; Song et al., 2007; Shilling et al., 2009; Huffman et al., 2009; Poulain et al., 2010; Massoli et al., 2010; Kuwata et al., 2011; Tritscher et al., 2011; Frosch et al., 2011). These studies reported O:C ratios from around 0.3 to 1. The higher O:C ratios were determined by Massoli et al. (2010) for extended periods of chemical aging; most of the average O:C ratios ranged from 0.3 to 0.5. The use of the O:C ratio can allow for easier classification of OA into different classes using the two-dimensional volatility basis set (2D-VBS, Donahue et al., 2011). Poulain et al. (2010) observed that the most oxygenated compounds were less volatile than the less oxygenated ones and Kuwata et al. (2011) found that the O:C ratio for α-pinene ozonolysis SOA depended on the mass concentration.

In addition to hygroscopicity and oxidation level, there have been a number of studies focusing on the volatility of SOA from the ozonolysis of α-pinene with either a thermodenuder (TD) (An et al., 2007; Kostenidou et al., 2009; Huffman et al., 2009; Poulain et al., 2010; Lee et al., 2011; Cappa and Wilson, 2011; Kuwata et al., 2011) or a volatility tandem differential mobility analyzer (V-TDMA) (Stanier et al., 2007; Jonsson et al., 2007; Tritscher et al., 2011). The difficulty in comparing these volatility studies stems from the different experimental setups used. Different heating methods (TD, V-TDMA, etc.), residence times in heating sections, and temperatures make it challenging to determine a common thread. For example, Poulain et al. (2010) observed nearly all of the SOA evaporated at 200°C at a residence time of 9 s in the heating section of a TD, but Lee et al. (2011) reported that most of the SOA evaporated at around 90°C for a residence time of 16 s. The use of the 1D-VBS (Donahue et al., 2006) can help overcome this obstacle, making it easier to compare volatility distributions rather than MFR values which are influenced by several different experimental factors (particle size, residence time in heating section, OA concentration, etc.) (Cappa, 2010; Riipinen et al., 2010; Kuwata et al., 2011).

Several studies have attempted to relate two of the three properties, but few have attempted to relate all three. Jimenez et al. (2009) proposed that hygroscopicity generally increases with the oxidation level expressed by the O:C ratio and that there is also an inverse relationship between the O:C ratio and volatility. Tritscher et al. (2011) used a volatility and hygroscopicity tandem differential mobility analyzer (V/H-TDMA) and an Aerodyne High





Resolution Time-of-Flight Aerosol Mass Spectrometer (HR-ToF-AMS, hereafter AMS) during the chemical aging of α-pinene SOA and found that volatility decreased while hygroscopicity and the O:C ratio remained fairly constant. Cerully et al. (2015) used a TD followed by a CCN counter, a scanning mobility particle sizer (SMPS), and an AMS in parallel and observed small

changes in hygroscopicity for ambient OA components with dramatically different volatilities and concluded that the more volatile compounds were more hygroscopic than the remaining material. Several other studies (Poulain et al., 2010; Kuwata et al., 2011; Hong et al., 2014; Hildebrandt Ruiz et al., 2015) have investigated the effects of environmental parameters on one or all of these properties. However, these results are often inconclusive or even

contradictory and the links among these three properties are yet to be elucidated.

A theoretical framework (Nakao, 2017) has attempted to relate these three properties using the 2D-VBS framework (Donahue et al., 2011). The approach utilized correlations between the O:C ratio, volatility, and hygroscopicity to predict lines of constant κ in the 2D-VBS. The study concluded that relatively volatile OA components with a low O:C ratio can

have the same hygroscopicity as OA with lower volatility and a higher O:C ratio.

The purpose of this work is the development of a method for the synchronous measurement of OA hygroscopicity, oxidation level, and volatility. In the next section, we describe the proposed technique that utilizes a suite of aerosol instrumentation to measure these properties. Then, the method is tested with α-pinene ozonolysis SOA. This SOA system has

20 been studied extensively so it can be a useful first test for the proposed experimental approach. Our objective is not to perform a comprehensive study of the properties of this SOA (which depend on SOA levels, relative humidity, etc.), but rather to use it as a pilot study. Finally, a data analysis technique is developed to interpret and synthesize the corresponding measurements.

**2 Methodology**

**2.1 Instrument setup**

A schematic of the proposed experimental setup can be seen in Fig. 1. Particles are sampled through either a TD or a by-pass (BP) line and then the sample stream is split between an AMS and a differential mobility analyzer (DMA, TSI, model 3081). The stream from the

30 DMA is split again between a condensation particle counter (CPC, TSI, model 3010/72) and a CCN counter (CCNC, Droplet Measurement Technologies). In this study, the AMS used a flow rate of 0.1 L min$^{-1}$ while the CPC and the CCNC used 0.3 and 0.5 L min$^{-1}$ respectively. The





sheath flow in the DMA was set to 8 L    min⁻¹ to allow for a 10:1 sheath to aerosol flow ratio as the particles were classified. The upscan of the DMA was set to 120 s and the downscan was set to 15 s.

## 2.2 Hygroscopicity

Hygroscopicity measurements were made with a CCNC, which generates supersaturations by exploiting water's higher diffusivity than heat in air (Roberts and Nenes, 2005). Its fast response time allows it to be coupled to a SMPS, a technique called Scanning Mobility CCN Analysis (SMCA, Moore et al., 2010). In our experiments, polydisperse aerosol was charged with a Po-210 neutralizer and then entered a DMA where the particles were

classified by their electrical mobility and counted by a CPC and the CCNC as the DMA voltage was scanned. The DMA downscan, since it is short and generates a large response in both the CPC and CCNC, was used to align both instruments' responses. An activation curve was produced by inverting the time series to generate number and CCN size distributions and dividing the CCN by the total particle concentration (CN) at each size. The activation diameter

was calculated by fitting the activation curve to a sigmoidal function:

$$\frac{\text{CCN}}{\text{CN}} = \frac{B}{\left(1+\frac{D_d}{D_{p50}}\right)^c} \tag{1}$$

where CCN/CN is the fraction of activated particles, $B$ and $c$ describe the asymptote and slope of the sigmoid respectively, $D_d$ is the dry diameter, and $D_{p50}$ is the diameter at which 50 % of the particles activate for a symmetric size distribution; this corresponds to the activation

diameter.

The method to determine κ follows the analysis done by Petters and Kreidenweis (2007) and will be explained briefly here. The defining equation is known as the κ-Köhler equation:

$$S = \frac{D^3 - D_d^3}{D^3 - D_d^3(1-\kappa)} \exp\left(\frac{4\sigma M_w}{RT\rho_w D}\right) \tag{2}$$

where $S$ is the saturation ratio, $D$ is the wet particle diameter, $D_d$ is the dry diameter, $\sigma$ is the

surface tension of the solution/air interface, $M_w$ is the molecular weight of water, $R$ is the universal gas constant, $T$ is the temperature, and $\rho_w$ is the density of water. For a selected dry diameter and κ, the critical supersaturation, $S_c$, can be computed from the maximum of Eq. (2). Then, lines of constant κ can be obtained by plotting $\log(S_c)$ as a function of $\log(D_d)$ and by adding experimentally-determined activation diameters at different supersaturations, the

corresponding κ can be estimated.



### 2.3 Oxygen content

An AMS was used to monitor the aerosol's composition. In our experiments, the AMS was operated in the higher sensitivity V-mode (DeCarlo et al., 2006) and used an averaging time of one minute. The data were analyzed in Igor Pro 6.22A (Wavemetrics) using "Squirrel" version 1.56D for unit mass resolution analysis and "Pika" version 1.15D for high resolution analysis. The O:C ratios reported here were calculated using the Canagaratna et al. (2015) method.

### 2.4 Volatility

Volatility measurements were made with the TD and the SMPS. These instruments were used to generate thermograms (MFR as a function of TD temperature). The MFR represents the fraction of particle mass that did not evaporate in the TD. The thermogram can be combined with a TD model (Riipinen et al., 2010), which describes the multicomponent OA evaporation to calculate the OA volatility distribution. The fitting algorithm has been described and evaluated by Karnezi et al. (2014). The 1D-VBS (Donahue et al., 2006) is used here, which discretizes the volatility distribution into logarithmically spaced bins based on an effective saturation concentration, $C^*$.

The TD used for this study consisted of two parts: a heating section and a cooling section. The heating section is 2 ft long and is surrounded by heating tape to control the temperature. The cooling section is also 2 ft long and contains activated carbon to avoid any recondensation while the aerosol returns to room temperature. Aerosol passes through the entire TD via 1.5 in tubing. Therefore, accounting for all the flows after the TD, the centerline residence time at 298 K in the heating section was 23 s.

As particles pass through the TD, some of the mass will evaporate, but some particles will also be lost to the walls. To characterize these losses, NaCl particles were generated and passed through the BP and TD alternately for several temperatures. By comparing the size distributions through the BP and TD, the particle losses were quantified as a function of temperature and particle size for the flowrate used in our experiments.

### 2.5 Smog chamber setup

All experiments were conducted in the Carnegie Mellon University smog chamber, a 10 m³ Teflon (Welch Fluorocarbons) reactor suspended in a temperature-controlled room (Pathak et al., 2007). Before each experiment, the chamber was flushed overnight with purified air under UV illumination (GE, model 10526 and 10244) to remove any potential contaminants.



Purified air was generated by passing compressed air through a high-efficiency particulate air (HEPA) filter to remove any particles, an activated carbon filter to remove any vapors, and silica gel to maintain the relative humidity (RH) at less than 5 %.

## 3 System test with ammonium sulfate aerosol

To determine whether the proposed and rather complex setup of several instruments operating together in series and in parallel was operating properly, a 1 g  L$^{-1}$ solution of ammonium sulfate was pumped through an atomizer (TSI, model 3075) at a constant rate of 90 mL h$^{-1}$ using a constant output syringe pump (Braintree Scientific, model BS-300). Before entering the chamber, the resulting droplets passed through a silica gel dryer to produce dry particles. Ammonium sulfate particles have traditionally been used in TD and CCN tests because they are easy to produce and behave as non-volatile at temperatures below approximately 150°C (Clarke, 1991; An et al., 2007). Furthermore, they are extremely hygroscopic with well-known properties. Their CCN activity has been shown to be consistent with Köhler theory (Cruz and Pandis, 1997). At different TD temperatures below 150°C, ammonium sulfate particles should behave the same hygroscopically as the particles passing through the BP. Our system uses the CCNC to measure the fraction of particles that will activate and become cloud droplets as a function of particle size and water vapor supersaturation, resulting in an activation curve.

The thermogram for the ammonium sulfate particles is shown in Fig. 2a. No aerosol mass evaporated at 50 and 100°C, but nearly all of the mass evaporated at 150°C. The remaining material at 150°C also included impurities in the ammonium sulfate solution that do not evaporate at that temperature. The measured activation diameter for the BP, two TD temperatures (50 and 100°C), and Köhler theory from this experiment is shown in Fig. 2b. The activation diameters at all supersaturations through the BP and the TD agreed with theory, confirming that our system operates properly for at least this simple model system.

## 4 Application to α-pinene ozonolysis SOA

The proposed experimental approach was then applied to SOA. For these experiments, α-pinene (Sigma-Aldrich, ≥ 99 %) was injected into the chamber using a heated septum injector with purified air as carrier flow. Ozone was generated by an ozone generator (AZCO, model HTU-500ACPS) and injected into the chamber after the α-pinene injection. Table 1 displays the different experimental conditions examined in this pilot study. The initial ozone





concentration and RH remained almost the same in all experiments, but the initial α-pinene concentration and water supersaturation in the CCNC were varied.

For Experiment 1, 100 ppb of α-pinene were injected into the chamber immediately followed by around 500 ppb of ozone. After one hour of reaction time, the ozonolysis was

practically complete and particles were sampled through the TD and BP alternatively for five temperatures (25, 50, 75, 100, and 125°C). The SOA mass concentration, measured with the SMPS assuming a density of 1.4 g cm$^{-3}$ (Kostenidou et al., 2007), increased immediately following the ozone injection and reached a maximum of around 108 μg m$^{-3}$. The SOA then began decreasing due to particles being lost to the chamber walls.

The thermogram, TD model prediction, and corresponding estimated volatility distribution, using the Karnezi et al. (2014) algorithm, for Experiment 1 are shown in Fig. 3. The TD model reproduced the MFR measurements well. For this experiment with OA around 100 μg m$^{-3}$, over two-thirds of the SOA had a $C^* = 1$ or 10 μg m$^{-3}$, 20% had a $C^* = 0.1$ μg m$^{-3}$, and 10 % had a $C^* = 0.01$ μg m$^{-3}$. The $C^* = 0.01$ μg m$^{-3}$ bin also includes compounds with even

lower volatilities. An effective enthalpy of vaporization, $\Delta H_{vap}$, equal to 65 kJ mol$^{-1}$ was estimated assuming an accommodation coefficient equal to unity.

Our TD results are comparable to those in the literature for α-pinene ozonolysis SOA. At 100°C, the MFR for the SOA in Experiment 1 was 0.29 ± 0.01. Huffman et al. (2009) and Poulain et al. (2010) reported a MFR of around 0.35 at 100°C, but their SOA levels were several

20  times larger (4–6) and their residence times were about half of the one used in this study. Kuwata et al. (2011) observed a MFR of 0.50 at a TD temperature of 100°C for lower SOA levels (25–37 μg m$^{-3}$) and a significantly shorter residence time (0.4 s). This further reiterates the difficulty in comparing volatility studies using different experimental methods. Generating reproducible TD results requires the same TD operated at the same flow rate and temperatures,

which is why our study uses the 1D-VBS, allowing for comparisons between studies, regardless of TD operating conditions.

In addition to volatility, the SOA's oxygen content was also measured. Figure 4 shows the average O:C ratio through the BP and TD at different temperatures for Experiment 1. The O:C ratio started around 0.49 and decreased at higher temperatures, ending at 0.39 ± 0.02 while

passing through the TD at 125°C. All of the O:C ratios at a TD temperature of 50°C and above were statistically lower than the O:C ratios through the BP and TD at 25°C (one-tailed $t$-test, $p < 0.0001$). The final O:C ratio of 0.39 corresponds to 11 % of the least volatile SOA (Fig. 3a). This indicates that the least volatile material in these experiments contained components that were not very oxidized at least on average.



Our O:C ratios fall into the reported range of O:C ratios in the literature for α-pinene ozonolysis SOA (Huffman et al., 2009; Massoli et al., 2010; Kuwata et al., 2011; Tritscher et al., 2011). Kuwata et al. (2011) observed that the O:C ratio dropped sometimes due to high TD temperatures, but argued that increases in the mass concentration were the main cause of the O:C ratio decreasing. Poulain et al. (2010) reported that the more oxygenated compounds were less volatile than the less oxygenated ones, which contradicts our results, but their conclusions were based on changing the RH at which the SOA was formed, which could impact the SOA's response to heat treatment.

The SOA's hygroscopic activity was also measured at the same time. The measured activation diameter as a function of temperature for Experiment 1 is shown in Fig. 5a. The activation diameter of all the SOA remained fairly constant around 140 nm during the experiment and as the temperature in the TD increased to 50°C. However, for even higher TD temperatures, the activation diameter increased, ending at $155 \pm 1$ nm for the least volatile 11 % of the SOA. The activation diameter through the TD at temperatures of 75°C and greater were statistically larger than the activation diameter through either the BP or TD at 25°C (one-tailed $t$-test, $p < 0.0001$). This indicates that the least volatile components of the α-pinene ozonolysis SOA also contained components that were not very hygroscopic. This is contradictory to conventional thinking, which assumes that the least volatile material is usually the most processed and therefore the most hygroscopic (Jimenez et al., 2009). However, the least volatile material in all experiments had consistently the lowest O:C ratio and highest activation diameter, indicating a more complex relationship between hygroscopicity, oxidation level, and volatility in this system.

The estimated κ parameters as a function of TD temperature in Experiment 1 can be seen in Fig. 5b. Similar to the activation diameters, the κ through the TD at temperatures greater than or equal to 75°C were statistically lower than the κ through either the BP or TD at 25°C (one-tailed $t$-test, $p < 0.0001$), reiterating the notion that the least volatile SOA contained components that were less hygroscopic than the rest of the SOA. The κ's obtained in all experiments were similar to observed values for α-pinene ozonolysis SOA in other CCNC studies (Engelhart et al., 2008; Massoli et al., 2010; Frosch et al., 2011) and the decrease in hygroscopicity after heat treatment has also been reported by Kuwata et al. (2011).

Detailed results for Experiments 2, 3, and 4 can be found in the supplement to this paper. They were all conducted with 50 ppb of α-pinene and around 500 ppb of ozone, generating around 40 μg m⁻³ of SOA. In general, the O:C ratios and κ's were slightly higher than those in Experiment 1. However, as with Experiment 1, these experiments also resulted in





statistically lower O:C ratios and κ's through the TD at higher temperatures. The following section proposes and discusses a novel analysis method to further investigate this behavior.

**5 Relating hygroscopicity and O:C with volatility**

The above data can be used to estimate the SOA's hygroscopicity and oxidation level as a function of volatility. The compounds in a volatility bin, $i$, have an average O:C ratio, $[O:C]_i$, and hygroscopicity parameter, $\kappa_i$. The O:C ratio and κ distributions as a function of volatility can be determined utilizing the data obtained at each TD temperature by using the following equations:

$$[O:C]_{TD} = \sum_{i=1}^{n} x_{i,TD} [O:C]_i \qquad (3)$$

$$\kappa_{TD} = \sum_{i=1}^{n} x_{i,TD} \kappa_i \qquad (4)$$

where $[O:C]_{TD}$ and $\kappa_{TD}$ are the measured O:C ratio and κ at a TD temperature, $x_{i,TD}$ is the SOA mass fraction in the $i^{th}$ bin at the same temperature, and $[O:C]_i$ and $\kappa_i$ are the unknown O:C ratio and κ for the $i^{th}$ bin. The mass fraction for each bin as a function of TD temperature is estimated by the TD model. An example for Experiment 1 is shown in Fig. 6. As expected, as the temperature in the TD increases, the more volatile components evaporate and the SOA is comprised of mostly low volatility compounds.

Equation (3) assumes implicitly that the SOA in the various volatility bins has similar average number of carbon atoms per molecule and also similar average molecular weights. These are clearly zeroth order approximations and introduce corresponding uncertainties in the estimated O:C ratio for each bin. Equation (4) is based on the work of Petters and Kreidenweis (2007). These equations can be used to generate a system of equations for both the O:C ratios and κ's that can be solved separately.

First we focus on how to determine the O:C ratio distribution as a function of volatility, but the process to determine the κ distribution is exactly the same. Each measurement at a specific thermodenuder temperature results to one equation of the form of Eq. 3 with unknowns the O:C ratios of the various volatility bins. For example, for Experiment 1, we used the average O:C ratios from Fig. 4 and the mass fractions from Fig. 6 at each TD temperature to generate 5 equations with 4 unknowns. The combination of the results of all experiments led to 18 equations (5 from Experiment 1, 4 from Experiment 2, etc.) with 4 unknown O:C ratios ($[O:C]_{0.01}$, $[O:C]_{0.1}$, etc.). The optimum values of the O:C ratios were determined by minimizing the squared residual between the measured ($[O:C]_{TD}$) and predicted ($\sum_{i=1}^{n} x_{i,TD} [O:C]_i$) O:C ratios. Matlab's linear least squares solver, *lscov*, was used for this task





The results for the O:C ratio distribution can be seen in Fig. 7a. The $C^* = 0.01$ µg m$^{-3}$ bin had the lowest O:C ratio, while the $C^* = 1$ µg m$^{-3}$ bin had the highest values. The $C^* = 0.1$ and 10 µg m$^{-3}$ bins had nearly identical O:C ratios. These results suggest that for this system and the conditions examined the relationship between O:C and effective volatility was not

monotonic.

The hygroscopicity parameter κ distribution was determined in exactly the same way and the results are depicted in Fig. 7b. The O:C ratios and corresponding κ's for each volatility bin were correlated extremely well ($r^2 > 0.99$, Fig. S13), which has been reported in numerous studies (Rickards et al., 2013). The SOA in the $C^* = 1$ µg m$^{-3}$ bin had the highest hygroscopicity

while the most and least volatile components had lower κ values.

In order to determine the robustness of our solution, we compared the predicted O:C ratios and κ's at each TD temperature using the best-fit values of the $[O:C]_i$ and $\kappa_i$ for each bin $i$ to the measured values for every experiment. For the predicted values, we used the distributions in Fig. 7, multiplied them by their corresponding mass fractions, and summed the

products up at each TD temperature. The predicted versus measured O:C ratios and κ's can be seen in Figs. 8 and 9 respectively. The O:C ratio distribution appears to be a better predictor than the κ distribution because most of the predicted O:C ratios lie on the 1:1 line or very close to it, while the predicted κ's were more scattered. However, essentially all of the predicted values were very close to the measured ones and the remaining values were within one standard

deviation of the measured values.

Figure 7 presents results that could, at least in principle, connect different, or even contradicting, results from previous studies. For example, Jimenez et al. (2009) proposed that hygroscopicity and the O:C ratio increase as volatility decreases. However, Cerully et al. (2015) reported contradicting results that the more volatile components of ambient OA were

more hygroscopic than the remaining material. The results presented here, albeit over a small range and only for α-pinene ozonolysis SOA, provide a context in which both conclusions can be true. As the volatility decreased from a $C^* = 10$ to 1 µg m$^{-3}$, the O:C ratio and κ increased (i.e. supporting the results of Jimenez et al. (2009)), but the O:C ratio and κ of the more volatile $C^* = 1$ µg m$^{-3}$ bin were higher than those of the $C^* = 0.01$ and 0.1 µg m$^{-3}$ bins (i.e. supporting

the results of Cerully et al. (2015)). Therefore, the approach described in this study can provide a more comprehensive method to determine the relationship between OA hygroscopicity, oxidation level, and volatility.





### 5.1 Sensitivity analysis

As a test, we optimized the system of equations again, but this time we removed one equation so that when we optimized the solution, we were only using 17 equations. This test allowed us to determine if there was one measurement that significantly affected the results in Fig. 7. In all cases but one, the average values were nearly identical to the distributions from Fig. 7, demonstrating that one equation was not overly influencing the results. The only substantial deviation from Fig. 7 was observed for the $C^* = 10$ µg m$^{-3}$ bin when the κ equation through the TD at 25°C for Experiment 1 was omitted. When the system was optimized without this equation, the κ for this bin decreased from 0.10 to 0.05. Since the mass fraction of the $C^* = 10$ µg m$^{-3}$ bin for Experiment 1 was almost double the next highest mass fraction in any measurement, it holds significantly more information, and therefore, weight, in the optimization process and impacts the solution for this bin. However, even if the κ for this bin changed when the corresponding measurement was omitted, the change is consistent with the large uncertainty bars for the κ of the $C^* = 10$ µg m$^{-3}$ bin in Fig. 7. This exercise suggest that our results appear to be relatively robust.

As seen in Fig. 9, the κ distribution in Fig. 7 under predicts all of the measured κ's from Experiment 2. In order to investigate the cause of this discrepancy, we used Eq. (4) to determine the κ distribution just for Experiment 2. Since this experiment only had 4 TD temperatures (Fig. S1), 4 equations could be written with 4 unknowns. However, when the system of equations was solved, the solution provided reasonable κ's for the three lowest bins, but produced a κ = -0.15 for the $C^* = 10$ µg m$^{-3}$ bin. We hypothesized that the unreasonable value for the highest bin was due to the small mass fraction of the SOA in this bin at all temperatures (Fig. S4). At 25°C, only 4 % of the SOA had a $C^* = 10$ µg m$^{-3}$ and those compounds completely evaporated as the temperature in the TD increased. Therefore, there was insufficient information to accurately estimate the properties of this bin for this experiment. To determine if this was true, we used Eq. (3) to solve for the O:C ratio distribution for Experiment 2. When we solved the system of equations, the solution again provided reasonable O:C ratios for the three lowest bins, but produced an O:C = 2.16 for the $C^* = 10$ µg m$^{-3}$ bin. Since both solutions provided unreasonable values for the highest bin, we concluded that the method is unable to accurately estimate the properties of bins with very low concentrations. However, when experiments with different concentrations were combined, as we did with Experiments 1–4 above, the method is able to determine reasonable estimates for the bins' properties.





To address the uncertainty introduced by the low concentration of material in the $C^* = 10$ µg m$^{-3}$ bin for Experiment 2, we solved the system of equations using the measurements at TD temperatures greater than or equal to 50°C and the three lowest bins ($C^* = 0.01$, 0.1, and 1 µg m$^{-3}$). This approach implicitly assumes that the material in the $C^* = 10$ µg m$^{-3}$ bin has evaporated at 50°C, which is reasonable because the mass fraction for the bin is less than 2 % at 50°C (Fig. S4). The results for the κ distribution can be seen in Fig. 10. When compared to the κ distribution in Fig. 7b, the κ's for the $C^* = 0.1$ and 1 µg m$^{-3}$ bins are the same within uncertainty, but the κ for $C^* = 0.01$ µg m$^{-3}$ bin is nearly twice that of the one in Fig 7b. This indicates that the SOA that had a $C^* = 0.01$ µg m$^{-3}$ in Experiment 2 was more hygroscopic than indicated in Fig. 7b. To demonstrate this, we predicted the κ's for Experiment 2 again. The κ's for the three highest bins remained the same, but we used the κ for the $C^* = 0.01$ µg m$^{-3}$ bin estimated in Fig. 10 instead of the one in Fig. 7b. The results can be seen in Fig. S14. With the larger κ for the $C^* = 0.01$ µg m$^{-3}$ bin, the predicted κ's were nearly identical to the measured ones. This demonstrates the variability that can occur from experiment to experiment, but the analysis method developed here can examine experiments separately to determine the cause of the variability.

## 6 Conclusions

This study evaluates the feasibility of a novel measurement and analysis technique to quantify OA's hygroscopicity, O:C ratio, and volatility distribution. The experimental approach used a CCNC to study hygroscopicity, an AMS to determine the O:C ratio, and a TD to evaluate volatility. The experimental setup was tested with SOA generated from the ozonolysis of α-pinene. The results of these experiments revealed that the O:C ratio and κ decreased as the SOA passed through the TD at higher temperatures. This indicates that the lowest volatility material in this system contained components that had lower O:C ratios and κ's.

An analysis technique to synthesize the data from this novel experimental setup was also developed. The results from this analysis confirmed that the SOA for this system had some low volatility material with a low O:C ratio and κ. It also showed that both low and high volatility compounds can have comparable oxidation levels and hygroscopicities.

This approach can be used to connect studies that were once thought to have contradicting results regarding the relationship between these three properties. Further chamber studies and ambient sampling are necessary in order to describe the relationship between OA





hygroscopicity, oxidation level, and volatility for multiple SOA systems, but also for ambient OA. This study was able to analyse a small portion of the 2D-VBS framework space, but this method can be utilized to help identify the relationship across the entire OA volatility axis. This approach also serves as an experimental branch to the theoretical framework proposed by

5 Nakao (2017).

*Acknowledgements.* This work was supported by the National Science Foundation grant 1455244.

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





**Table 1.** Description of experimental conditions used in this study.

| Exp. | α-Pinene (ppb) | Ozone (ppb) | RH (%) | Supersaturation (%)[a] | Max. OA (μg m$^{-3}$) |
|------|----------------|-------------|--------|------------------------|------------------------|
| 1 | 100 | ~500 | <15 | 0.20 | 108 |
| 2 | 50 | ~500 | <15 | 0.30 | 35 |
| 3 | 50 | ~500 | <15 | 0.25 | 39 |
| 4 | 50 | ~500 | <15 | 0.27 | 46 |

[a]CCNC supersaturation was held constant during experiments in order to allow sufficient time for an average activation diameter to be measured at each TD temperature.


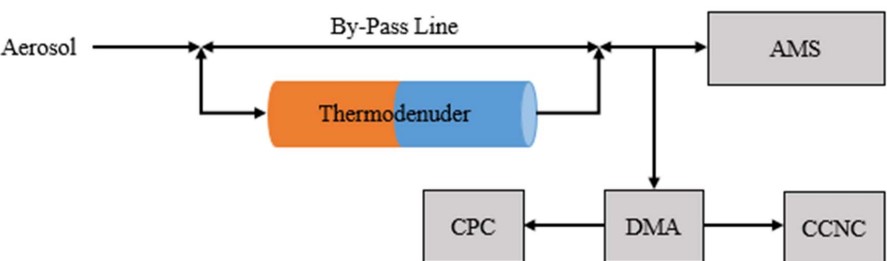

**Figure 1.** Schematic of the experimental setup for the hygroscopicity, oxidation level, and volatility measurements. The sampling technique employs a TD to measure volatility, an AMS to study the oxidation level, and a CCNC to determine hygroscopic activity.





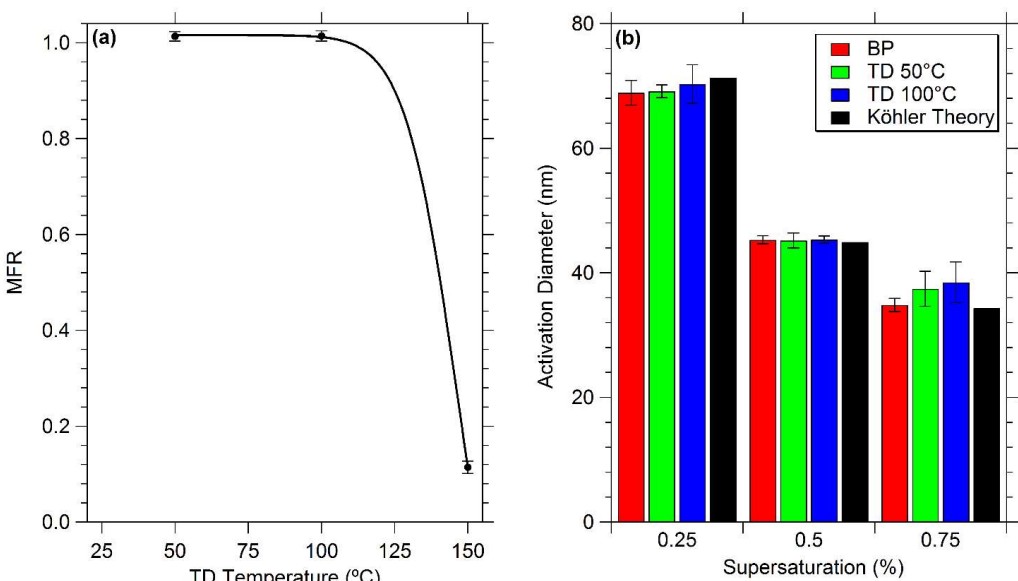

**Figure 2.** (a) The thermogram for ammonium sulfate aerosol. (b) The calculated activation diameter at three CCNC supersaturations for the BP (red), two TD temperatures (green and blue), and Köhler theory (black) for ammonium sulfate aerosol. The error bars represent one standard deviation of the mean.



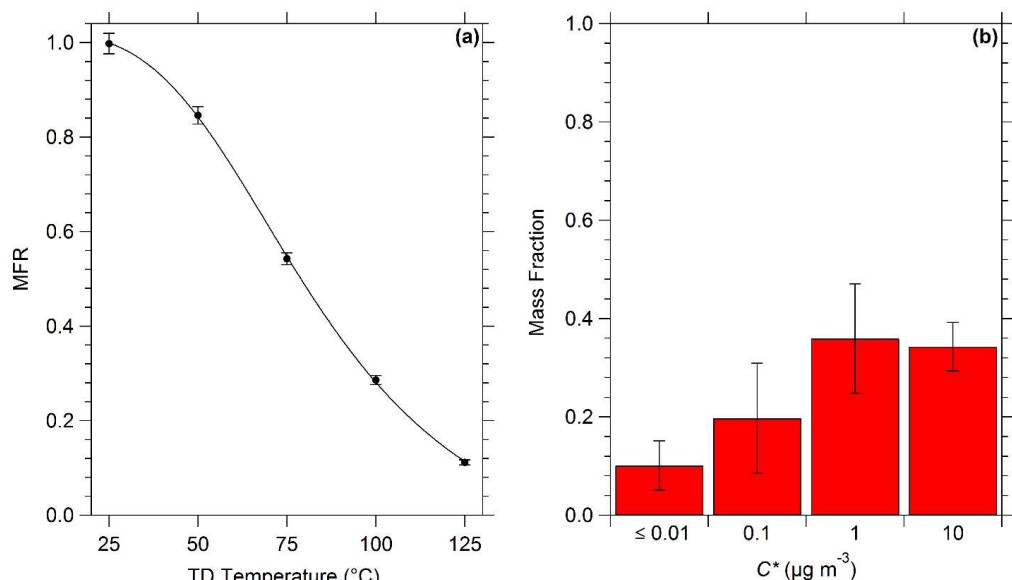

**Figure 3.** (a) Thermogram, corrected for losses in the TD, for Experiment 1 with the fit from the TD model (Riipinen et al., 2010; Karnezi et al., 2014). The error bars represent one standard deviation of the mean. (b) SOA volatility distribution for Experiment 1 using the 1D-VBS framework (Donahue et al., 2006). The error bars correspond to one standard deviation of the solution calculated by the model.





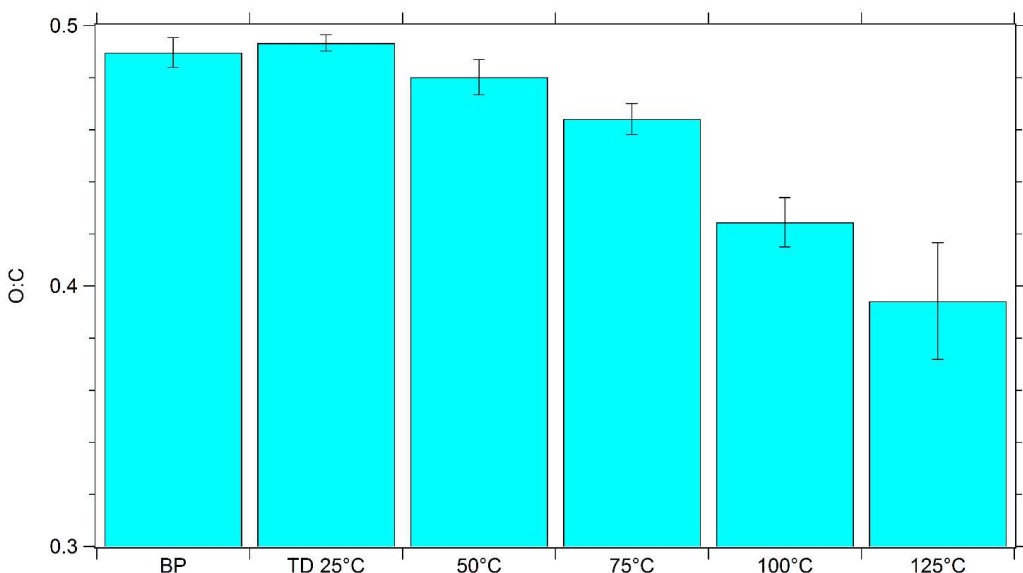

**Figure 4.** The average O:C ratio observed through the BP and several TD temperatures for Experiment 1. The error bars represent one standard deviation of the mean. The O:C ratios at a TD temperature of 50°C and greater were statistically smaller than the values at the BP and the TD at 25°C.





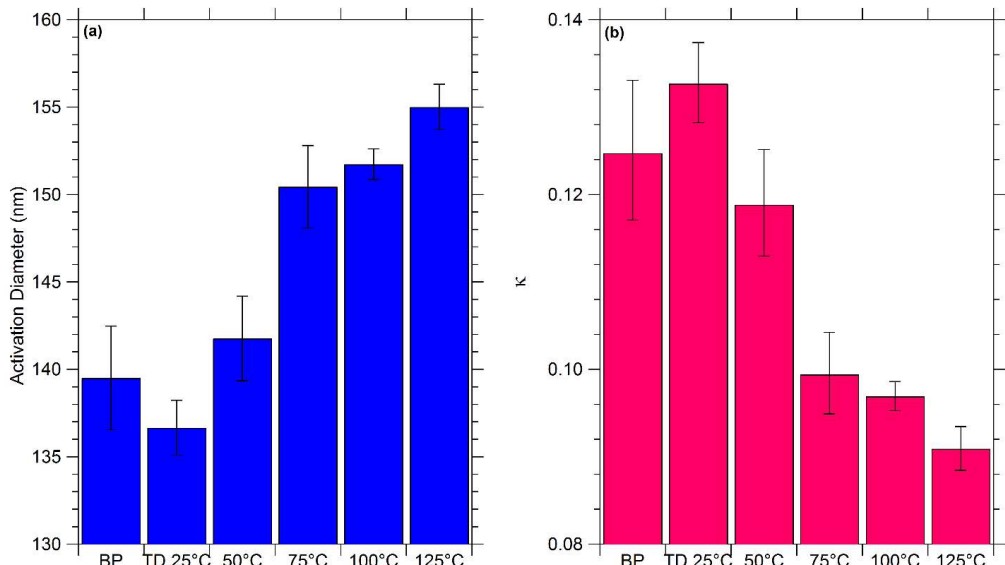

**Figure 5.** (a) The average activation diameter observed at 0.2 % supersaturation in the CCNC for Experiment 1. The error bars represent one standard deviation of the mean. (b) The estimated κ values for Experiment 1. The error bars were obtained by estimating the κ at +/- one standard deviation of the average activation diameter measured. The values at TD temperatures of 75°C and greater were statistically different from the values at the BP and the TD at 25°C.





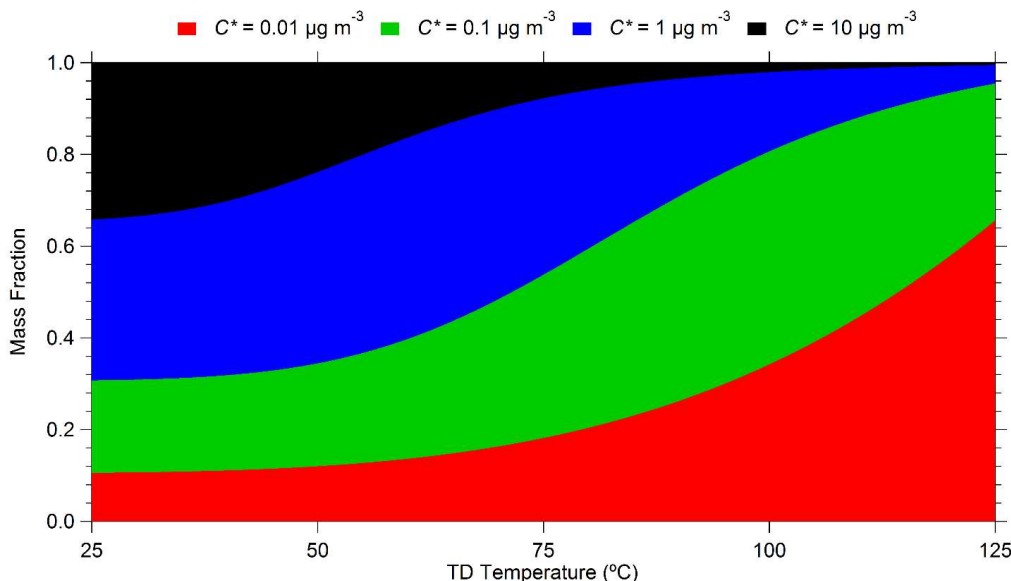

**Figure 6.** The estimated mass fractions for each volatility bin as a function of TD temperature for Experiment 1. Red represents the $C^* = 0.01$ μg m$^{-3}$ bin, green the $C^* = 0.1$ μg m$^{-3}$ bin, blue the $C^* = 1$ μg m$^{-3}$ bin, and black the $C^* = 10$ μg m$^{-3}$ bin.



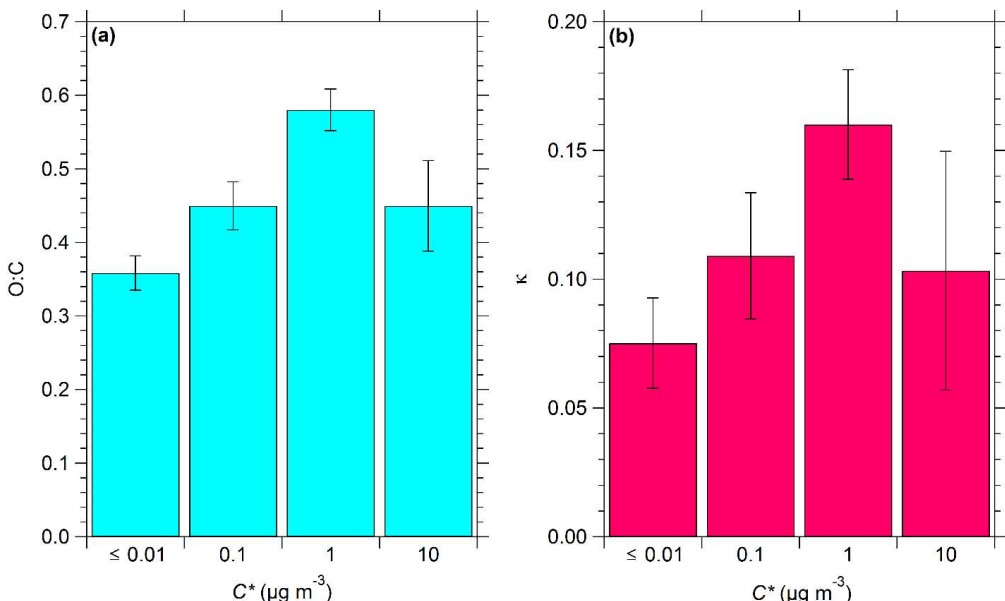

**Figure 7.** The (a) O:C ratio and (b) $\kappa$ distributions for the volatility bins characterized in this study for α-pinene ozonolysis SOA. The error bars represent one standard deviation of the mean obtained from the least squares solver.





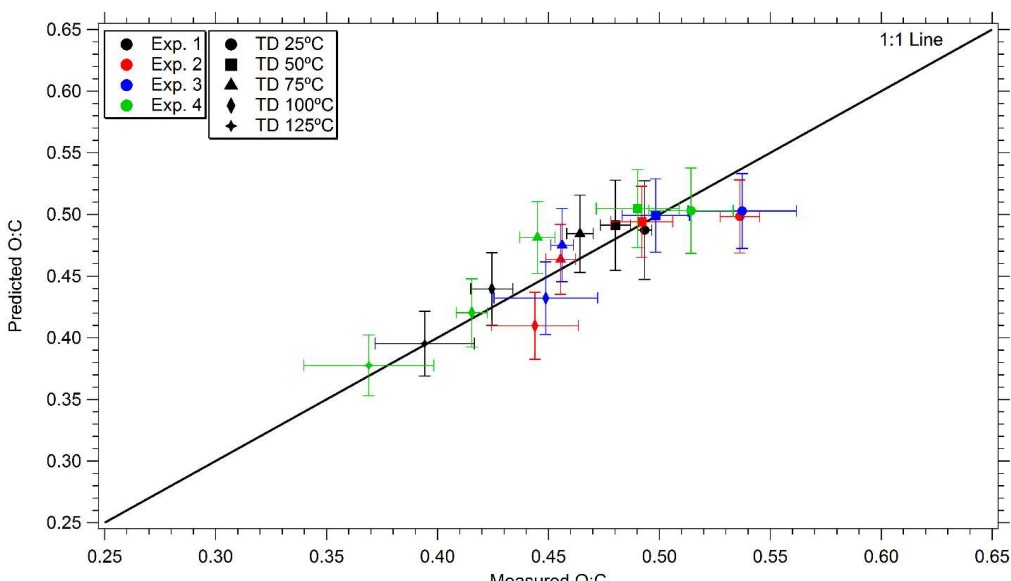

**Figure 8.** The predicted versus measured O:C ratios for all TD temperatures in all of the experiments. The color indicates the experiment number and the symbol indicates the TD temperature. The error bars for the predicted O:C ratios were obtained by predicting the O:C ratios using the O:C ratio distribution at +/- one standard deviation in Fig. 7a.

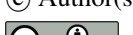



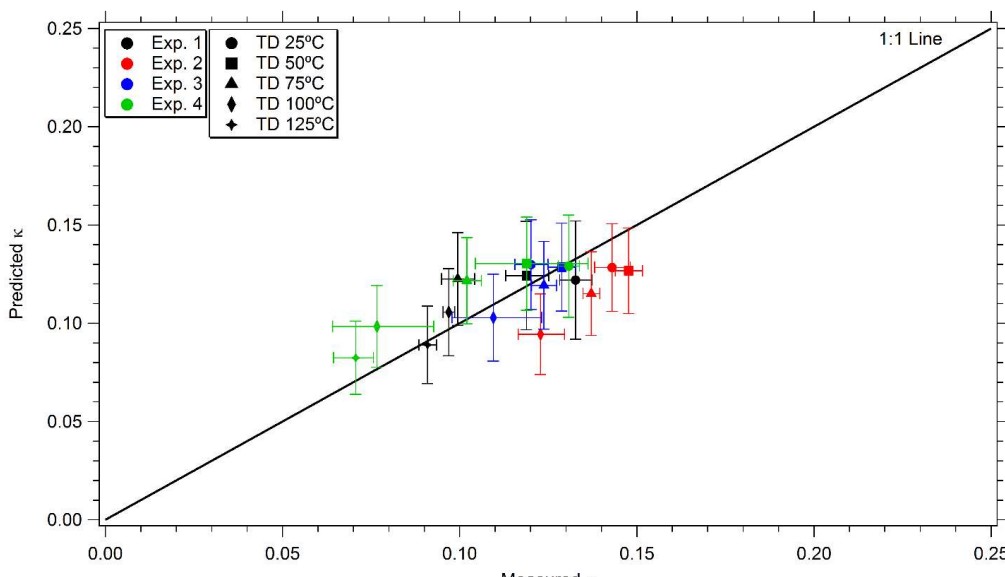

**Figure 9.** The predicted versus measured κ's for all TD temperatures in all of the experiments. The color indicates the experiment number and the symbol indicates the TD temperature. The error bars for the predicted κ's were obtained by predicting the κ's using the κ distribution at +/- one standard deviation in Fig. 7b.





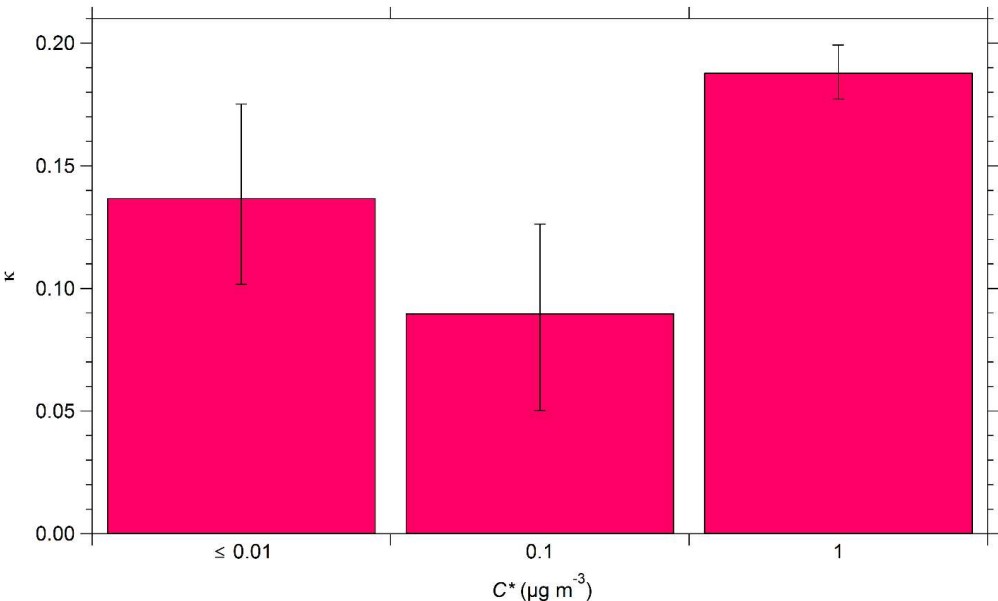

**Figure 10.** The κ distribution that resulted from solving Eq. (4) for TD temperatures greater than 50°C and the three lowest bins for Experiment 2. The error bars represent one standard deviation obtained by solving the equations at +/- one standard deviation of the measured κ.