# Peer review of "A technique for the measurement of organic aerosol hygroscopicity, oxidation level, and volatility distributions"

_Atmospheric Measurement Techniques, 2017_

## Referee Comment (RC1) · Anonymous Referee #2 · 24 Aug 2017

In this submission, the author combines several established aerosol instruments to measure hygroscopicity, volatility, and oxygen to carbon ratio simultaneously. A new inversion technique to handle the output from the instruments is used. This technique assigns hygroscopicity, and oxygen to carbon ratio to each volatility bin in  $log10(C^*)$  space. This requires several assumptions, and no support for the legitimacy of the assumptions are made, however, to do so would be difficult and likely beyond the scope of the submission. There appears to be enough information to reproduce the experiment as well as the inversion routine.

The conclusion is original and adds to the discussion of oxygen-to-carbon ratio and

hygroscopicity as a function of volatility bin. Several studies in this area contradict one another and sometimes theory. The paper's conclusion offers a possibility for previous results to be complimentary, but would require a shift in the proposed theory.

The author submits the results of 4 experimental conditions, and the paper uses a single experiment as an example. The results of other experiments appear to generally corroborate the author's conclusions and are located in the supplemental material. All experiments are used to test the noise and limitations of the proposed analysis method.

Several comments are listed below. In addition to those comments, a rewrite of the abstract should also be entertained as the first sentences are long and confusing. The main body of the document is clear and mostly precise. No large rewrites are deemed necessary, and I have no major objections.

Page 1 line 28: Acute not proper for all 3 cites. Miller is a long term study, Dockery is for acute aerosol episodes (not acute mortality), but Brooks seems appropriate.

Page 2 line 12: Soften statement. Authors studying cancer causing aerosols may disagree with the "three of the most important properties of organic aerosol."

Page 2 line 14-15: Hygroscopicity is the measure of a volume of water associated with a unit volume of solute. This is a bit more precise than "ability to absorb water." Hygroscopicity is not a measure of a particles ability to form cloud droplets.

Page 3 line 7-8: Massoli et al doesn't seem to make those conclusions. The Massoli paper cites Good et al 2010 and Petters et al 2009 which do make the proposed explanations.

Page 3 line 18-19: Insert comma before "and."

Page 3 line 26-29: These two sentences can be combined to reduce repetitive nature.

Page 5 line 14: Please specify which diffusivity. Mass diffusivity describes the movement of the mass, while thermal diffusivity describes the movement of energy. Page 5 line 15: Replace "its" with a proper noun to avoid confusion.

Page 5 line 15: replace "a" with "an"

Page 6 line 10: Confused by the word "adding." Does the author mean that we look up the activation diameter on the log(Sc)-log(Dd) plot? A few more words may be necessary.

Page 6 line 14: "higher sensitivity" may not be necessary. If used, should specify that the author desired a higher sensitivity to signal and not mass-to-charge.

Page 8 line5-7: The thermogram (figure 2a) displays the evaporation of Ammonium Sulfate below 150C, but page 7 line 28 says Ammonium Sulfate is involatile below 150C. In my experience, the thermogram is correct.

Page 8 line 10-12: Which theory? Kohler theory or previous observations. Appears Ammonium Sulfate is still stable at 100C, but disassociates at higher temperatures. Is the hygroscopicity of disassociated Ammonium Sulfate the same as pure Ammonium Sulfate? What about when including impurities? Are these in the "theory?"

Page 9 line 23-25: Some may contend that 0.39 is oxidized, especially when compared to alpha-pinene. A comparison may be better.

Page 11 line 18-19. In order to use equation 4, you must also assume density is constant between all volatility bins.

---

## Referee Comment (RC2) · F. Brechtel (Referee) · 27 Aug 2017

Review by Fred Brechtel

General comments:

A more appropriate title would be: A technique for the measurement of organic aerosol cloud nucleating potential, oxidation level, and volatility distributions. Hygroscopicity is more generally connected with measurements made below 100% RH using an HTDMA or some similar tool.

Why limit the method to only organic aerosol?

Please add a short discussion in the introduction section of the application of the technique to unknown composition ambient aerosol.

Techniques where multiple aerosol properties are determined simultaneously can be made much stronger by adding size resolution – e.g. by pre-selecting a time sequence of monodisperse particles with a DMA upstream of the various instruments, more easily interpretable data would likely result. This very likely, in part, explains the largely inconclusive previous studies as noted on page 4 line 15-20. Although beyond the scope of this work, I highly recommend the authors consider modifying their setup so a DMA is positioned upstream of the thermodenuder/bypass so all instruments sample the exact same monodisperse aerosol after having undergone the exact some thermal (or no) pre-treatment. This configuration will also help eliminate the ambiguity in the data comparison between aerodynamic diameter measured by the AMS and the electrical mobility diameter measured by the DMA and delivered to the CCN and CPC. Admittedly, the time resolution of the measurement will be sacrificed, but the much easier to interpret data and interesting size-dependent results that would be produced could be very exciting.

Consider the following thought experiment considering an unknown composition ambient aerosol. As configured, the AMS samples the polydisperse particle distribution after a certain temperature exposure in the TD. Any number of originally differently sized particles could contribute to the AMS response at a single aerodynamic diameter depending on the volatility distribution of the input aerosol to the TD. In fact, if some of the input aerosol to the TD were soot (or some other non-volatile species) coated with volatile SOA, the AMS may not detect the non-volatile core left after treatment in the TD. The scanning DMA and CCN systems would detect the presence of the non-volatile core, but there would likely not be a corresponding signal in the AMS. How would these results be interpreted? This situation is one where selecting monodisperse aerosol upstream of the TD would greatly facilitate interpretation of the measurement results.

Most volatility studies assume that material volatile at or below the thermodenuder temperature setpoint is evaporated to the gas phase. However, it seems that chemical reactions stimulated by the high temperature environment may also change the composition and oxidation state of the particles within the TD. Can you comment on the likelihood of this?

Another general question worth asking is: what is the relevance of volatility studies to the atmospheric aerosol since they are never exposed to most of the temperatures used in TD studies?

Finally, it is not very clear to me why volatility would necessarily be well connected to hygroscopicity.

More detailed comments:

Page 2, line 10: suggest adding that the gases can produce new particles AND condense on pre-existing particles.

Page 2, line 29: please clarify what 'this SOA system' refers to.

Page 3, line 34: is 'MFR' defined somewhere?

Page 4, line 4: suggest "...that the hygroscopicity of organic aerosol generally increases with ..."

Page 5, line 5: were flow mixers employed upstream of the two sample flow splits to ensure the AMS/DMA and CPC/CCN received the same aerosol populations?

Page 6, line 20: please explain how 'mass fraction remaining' in the TD is determined using SMPS measurements. How is the density measured to convert the mobility volume distribution to a mass distribution?

Page 7, line 4: it is worth mentioning here the shorter residence time you expect at your maximum TD operating temperature.

Page 7, line 4: please comment on the expected range in residence time within the TD. For the low flow rate and large tube diameter, do you expect laminar flow conditions? If so, wouldn't particles traveling near the centerline experience a residence time roughly half those near the walls? If this is the case, how would this impact your results?

Page 9, line 1: please explain what C* means

Page 9, line 15: I am confused by the statement that using the 1D-VBS allows for comparison of different TD studies regardless of TD operating conditions. If the physical residence time in a TD is too short to allow complete volatilization, or if the temperature-time history within a TD is not represented by the temperature setpoint of the study, how would a model allow successful intercomparisons of different TD study results?

Page 10, line 2: please add the supersaturation value set in the CCN instrument after activation diameter

Page 10, line 6: 155 +/- 1 nm? This level of uncertainty or standard deviation in the measured activation diameter is a little hard to believe, you are reporting a size variation of only +/-0.06%. If you were to scan monodisperse 155 nm diameter PSL calibration particles 50 times and calculate the variation in the measured peak size, my guess is you would see higher intrinsic measurement uncertainty than 0.06%.

Page 10, line 10: or that chemical changes to the particles that occurred within the TD rendered the particles less CCN active...?

Page 10, line 15: You make a very good point here that trying to generalize relationships between volatility, hygroscopicity/cloud activity and oxidation level is over simplifying the situation. However, it would be extremely useful if such relationships could be developed, perhaps by dividing organic species into certain families or structures with key similarities. Any comments?

Page 11, line 10: can you comment briefly on how you would extend this analytical approach to unknown composition ambient aerosol? E.g. mixtures of insoluble material,

soluble inorganics and soluble SOA?

Page 11, end of page: missing period

Page 13: in performing the sensitivity analysis – did you systematically eliminate each one of the 18 equations and run 17 studies? Or did you just run one study after picking a single equation to eliminate?

How would your sensitivity results change if you 'randomly' added + or – one standard deviation to the experimental data used to constrain the equations? What if you added one standard deviation to all of your data?

Page 13, line 17: suggests

Page 14, line 2: if the method has difficulty with low concentrations, does this impact its usefulness for studies of the ambient aerosol?

Page 14, it would be helpful if you rewrote the paragraph starting at line 6 so it was easier to understand.

---

## Referee Comment (RC3) · Anonymous Referee #1 · 8 Sep 2017

This manuscript describes an instrument configuration used to measure the volatilitydependent composition and CCN activity of an aerosol. The experimental approach was used to measure a well-characterized inorganic aerosol and a well-studied secondary organic aerosol produced from ozonolysis of alpha-pinene. The measurements are useful and the description would be of interest for some readers of this journal. But while there is nothing wrong with the manuscript, it describes a rather straightforward configuration of components used by many researchers and, thus, is not especially novel. To me, the preliminary results are the most interesting element of the manuscript. But I wonder whether it would make more sense to present them in a separate paper focused on the interpretation of the data and not the experimental technique. Nevertheless, the manuscript could be suitable for publication after the points below are addressed. The text is easily understood but would benefit from additional editing.

The observation that O:C and hygroscopicity decreased for the least volatile particles is certainly interesting. The authors provide a plausible explanation for the unexpected pattern. But further explanation of the experimental technique is required. Specifically, the time required for the full measurement sequence and the order of the TD temperatures should be provided. The rationale for this is that it seems possible that the aerosol continued to evolve during the measurements such that what is sampled when the TD is set at 100 C differs from that when it is at 150 C.

Page 6, line 22: Rather than just stating that the loss rate was determined it should be reported. A large correction for a high loss rate could significantly increase uncertainty in the measured MFR.

Minor points: Page 2, line 19: "...is reasonable CCN material using supersaturated conditions" should be re-worded.

Page 4, line 23: Why "proposed"?

Page 5, line 11: What is a "large response"?

Page 5, line 13: If this level of detail about the analysis of the data is going to be provided then the approach to inverting the SMPS-CPC and SMPS-CCNC distributions should be included.

Page 6, line 21: Use metric or change in to inch or to in. to clarify.

Page 7, line 11: "...behave as non-volatile" should be re-worded.

Page 7, line 12: Replace "extremely" with something like "very".

Page 7, line 16: This final sentence repeats what was already explained.
Page 8, line 24: I appreciate what you are trying to explain here, but as written the first and second parts of the sentence seem contradictory.

Page 9, line 18: "...conventional thinking, which assumes" should be re-worded.

Page 9, line 24: Change to something like "Similar to the pattern observed with the activation diameter..."

---

## Author Response (AR1)

**(1)** This manuscript describes an instrument configuration used to measure the volatility dependent composition and CCN activity of an aerosol. The experimental approach was used to measure a well-characterized inorganic aerosol and a well-studied secondary organic aerosol produced from ozonolysis of alpha-pinene. The measurements are useful and the description would be of interest for some readers of this journal. But while there is nothing wrong with the manuscript, it describes a rather straightforward configuration of components used by many researchers and, thus, is not especially novel. To me, the preliminary results are the most interesting element of the manuscript. But I wonder whether it would make more sense to present them in a separate paper focused on the interpretation of the data and not the experimental technique. Nevertheless, the manuscript could be suitable for publication after the points below are addressed. The text is easily understood but would benefit from additional editing.

We thank the referee for the helpful comments and suggestions. While we do agree that the experimental setup combines techniques that have been used in the past, we are proposing a new way to analyze the corresponding data and then synthesize them following the 2D-VBS framework. Our hope is that the same experimental method combined with the new analysis technique will provide valuable insights into these properties and their relationships. We do agree that the first results are quite interesting and this is the reason that we prefer to combine the presentation of the experimental technique, the data analysis method, and the results of this pilot study to demonstrate the utility of the proposed approach.

**(2)** The observation that O:C and hygroscopicity decreased for the least volatile particles is certainly interesting. The authors provide a plausible explanation for the unexpected pattern. But further explanation of the experimental technique is required. Specifically, the time required for the full measurement sequence and the order of the TD temperatures should be provided. The rationale for this is that it seems possible that the aerosol continued to evolve during the measurements such that what is sampled when the TD is set at 100 C differs from that when it is at 150 C.

We have added the requested details about the experimental technique. In short, the aerosol is passed through the TD for 7-10 SMPS scans (16-23 minutes). Then, the aerosol is sent through the by-pass line until the TD is at the next set point (usually 7-10 SMPS scans). Once the TD is at the desired new temperature, aerosol is sent through the TD for the same number of SMPS scans and this process is repeated until measurements in all TD temperatures have been performed. Depending on the number of temperature set points, a full sequence can take anywhere from 2.5-3.5 hours. In our experiments the temperature has been increasing during the TD scan. The potential evolution of the SOA during these measurements can be checked by comparing the AMS spectra to the original one in the beginning of the scan. The measurements can also be repeated by performing a second TD scan and collecting a second thermogram. In our experiment the change of the SOA spectra during the experiment was minimal (less than 5 degrees) in all cases. This information has been added to the revised paper.

**(3)** Page 6, line 22: Rather than just stating that the loss rate was determined it should be reported. A large correction for a high loss rate could significantly increase uncertainty in the measured MFR.

This is a good point. We have included the corresponding information about the TD loss rate constant as a function of temperature and particle size in the Supplement.

Minor points

**(4)** Page 2, line 19: ". . .is reasonable CCN material using supersaturated conditions" should be re-worded.

We have removed "using supersaturated conditions" from this sentence to avoid confusion. In the following sentence we added that all of the studies cited there performed measurements in supersaturated conditions.

**(5)** Page 4, line 23: Why "proposed"?

We included the word proposed because we are proposing a method to measure and relate these three properties. To avoid confusion, we have removed this word.

**(6)** Page 5, line 11: What is a "large response"?

We have removed this relative term and rewrote the sentence to state that we aligned the size distributions using the minimum that occurs between the DMA upscan and downscan.

**(7)** Page 5, line 13: If this level of detail about the analysis of the data is going to be provided then the approach to inverting the SMPS-CPC and SMPS-CCNC distributions should be included.

We have rewritten this section to include more detail regarding the inversion technique and included a reference to the study that developed this method.

**(8)** Page 6, line 21: Use metric or change in to inch or to in. to clarify.

Done.

**(9)** Page 7, line 11: ". . .behave as non-volatile" should be re-worded.

We have rewritten this sentence.

**(10)** Page 7, line 12: Replace "extremely" with something like "very".

Done.

**(11)** Page 7, line 16: This final sentence repeats what was already explained.

We have removed this sentence to avoid repetition.

**(12)** Page 8, line 24: I appreciate what you are trying to explain here, but as written the first and second parts of the sentence seem contradictory.

We have removed the beginning of this sentence and added the rest to the end of the previous sentence to avoid repetition and confusion.

**(13)** Page 9, line 18: ". . .conventional thinking, which assumes" should be re-worded.

Done.

**(14)** Page 9, line 24: Change to something like "Similar to the pattern observed with the activation diameter. . ."

We have made the recommended change.

**Anonymous Referee #2**

**(1)** In this submission, the author combines several established aerosol instruments to measure hygroscopicity, volatility, and oxygen to carbon ratio simultaneously. A new inversion technique to handle the output from the instruments is used. This technique assigns hygroscopicity, and oxygen to carbon ratio to each volatility bin in $\log 10(C^*)$ space. This requires several assumptions, and no support for the legitimacy of the assumptions are made, however, to do so would be difficult and likely beyond the scope of the submission. There appears to be enough information to reproduce the experiment as well as the inversion routine.

The conclusion is original and adds to the discussion of oxygen-to-carbon ratio and hygroscopicity as a function of volatility bin. Several studies in this area contradict one another and sometimes theory. The paper's conclusion offers a possibility for previous results to be complimentary, but would require a shift in the proposed theory.

The author submits the results of 4 experimental conditions, and the paper uses a single experiment as an example. The results of other experiments appear to generally corroborate the author's conclusions and are located in the supplemental material. All experiments are used to test the noise and limitations of the proposed analysis method.

Several comments are listed below. In addition to those comments, a rewrite of the abstract should also be entertained as the first sentences are long and confusing. The main body of the document is clear and mostly precise. No large rewrites are deemed necessary, and I have no major objections.

We appreciate the positive assessment of our work by the referee. We have rewritten the abstract shortening the initial sentences. Detailed responses to the comments can be found below.

**(2)** Page 1 line 28: Acute not proper for all 3 cites. Miller is a long term study, Dockery is for acute aerosol episodes (not acute mortality), but Brooks seems appropriate.

We have removed the two citations following the suggestion of the referee.

**(3)** Page 2 line 12: Soften statement. Authors studying cancer causing aerosols may disagree with the "three of the most important properties of organic aerosol."

We rewrote this sentence to state that these properties are important for the OA lifetime.

**(4)** Page 2 line 14-15: Hygroscopicity is the measure of a volume of water associated with a unit volume of solute. This is a bit more precise than "ability to absorb water." Hygroscopicity is not a measure of a particles ability to form cloud droplets.

We have revised this sentence to better define hygroscopicity.

**(5)** Page 3 line 7-8: Massoli et al doesn't seem to make those conclusions. The Massoli paper cites Good et al 2010 and Petters et al 2009 which do make the proposed explanations.

We have added these two citations and removed the Massoli et al. citation to better reflect the proposed explanations.

**(6)** Page 3 line 18-19: Insert comma before "and."

Done.

**(7)** Page 3 line 26-29: These two sentences can be combined to reduce repetitive nature.

We have combined the two sentences.

**(8)** Page 5 line 14: Please specify which diffusivity. Mass diffusivity describes the movement of the mass, while thermal diffusivity describes the movement of energy.

The CCNC exploits water's higher mass diffusivity than heat's thermal diffusivity in air. We have revised the manuscript to make this clear.

**(9)** Page 5 line 15: Replace "its" with a proper noun to avoid confusion.

Done.

**(10)** Page 5 line 15: replace "a" with "an"

Done.

**(11)** Page 6 line 10: Confused by the word "adding." Does the author mean that we look up the activation diameter on the log(Sc)-log(Dd) plot? A few more words may be necessary.

We have revised this sentence describing how we estimated $\kappa$ from the activation diameters. We plotted the activation diameters at their measured supersaturations on the $\log(S_c)$-$\log(D_d)$ plot and then $\kappa$'s were estimated by which line of constant $\kappa$ on this plot fit the activation diameters.

**(12)** Page 6 line 14: "higher sensitivity" may not be necessary. If used, should specify that the author desired a higher sensitivity to signal and not mass-to-charge.

We have removed "higher sensitivity" to avoid confusion.

**(13)** Page 8 line 5-7: The thermogram (Figure 2a) displays the evaporation of Ammonium Sulfate below 150 C, but page 7 line 28 says Ammonium Sulfate is involatile below 150 C. In my experience, the thermogram is correct.

The evaporation of ammonium sulfate particles in a TD expressed as mass fraction remaining depends not only on the TD temperature but also on particle size and residence time of the aerosol in the TD. We have changed the temperature in the manuscript to "100°C" to cover all these cases and to be consistent with Figure 2.

**(14)** Page 8 line 10-12: Which theory? Kohler theory or previous observations. Appears Ammonium Sulfate is still stable at 100C, but disassociates at higher temperatures. Is the hygroscopicity of disassociated Ammonium Sulfate the same as pure Ammonium Sulfate? What about when including impurities? Are these in the "theory?"

The activation diameters through the BP and the TD at 50 and 100°C are compared to Köhler theory assuming pure ammonium sulfate aerosol. We have added "Köhler" before theory to make this clearer. Disassociated ammonium sulfate will have the same hygroscopicity because the ammonium and sulfate ions will interact with water similarly. When aerosols have impurities the overall κ will be higher or lower depending on the κ's of the impurities due to the mixing rule.

**(15)** Page 9 line 23-25: Some may contend that 0.39 is oxidized, especially when compared to alpha-pinene. A comparison may be better.

We have rephrased this sentence. We now compare the O:C ratio through the BP to the O:C ratio through the TD at 125°C in order to show that the least volatile material is less oxidized than the overall SOA.

**(16)** Page 11 line 18-19. In order to use equation 4, you must also assume density is constant between all volatility bins.

We now state this assumption in the corresponding discussion.

**Fred Brechtel**

General comments:

**(1)** A more appropriate title would be: A technique for the measurement of organic aerosol cloud nucleating potential, oxidation level, and volatility distributions. Hygroscopicity is more generally connected with measurements made below 100% RH using an HTDMA or some similar tool.

We can see the point, but one could also argue exactly the opposite: measurements of CCN properties are better for the quantification of the hygroscopicity of a compound given the non-ideal solution effects often dominating behavior below 100% RH. Given that it is clear that the hygroscopic parameter kappa estimated in this method is $\kappa_{CCN}$ we would prefer to keep the same title. To avoid any confusion we clarify now both in the Abstract and the Conclusions that the hygroscopicity is based on CCN activity.

**(2)** Why limit the method to only organic aerosol?

The method can be easily applied to inorganic or mixed aerosol populations. However, given that the volatility and hygroscopicity of the major inorganic aerosol components in the atmosphere is well understood, the organic aerosol is the primary focus of this work.

**(3)** Please add a short discussion in the introduction section of the application of the technique to unknown composition ambient aerosol.

We have added the corresponding paragraph.

**(4)** Techniques where multiple aerosol properties are determined simultaneously can be made much stronger by adding size resolution – e.g. by pre-selecting a time sequence of monodisperse particles with a DMA upstream of the various instruments, more easily interpretable data would likely result. This very likely, in part, explains the largely inconclusive previous studies as noted on page 4 line 15-20. Although beyond the scope of this work, I highly recommend the authors consider modifying their setup so a DMA is positioned upstream of the thermodenuder/bypass so all instruments sample the exact same monodisperse aerosol after having undergone the exact some thermal (or no) pre-treatment. This configuration will also help eliminate the ambiguity in the data comparison between aerodynamic diameter measured by the AMS and the electrical mobility diameter measured by the DMA and delivered to the CCN and CPC. Admittedly, the time resolution of the measurement will be sacrificed, but the much easier to interpret data and interesting size-dependent results that would be produced could be very exciting.

Consider the following thought experiment considering an unknown composition ambient aerosol. As configured, the AMS samples the polydisperse particle distribution after a certain temperature exposure in the TD. Any number of originally differently sized particles could contribute to the AMS response at a single aerodynamic diameter depending on the volatility

distribution of the input aerosol to the TD. In fact, if some of the input aerosol to the TD were soot (or some other non-volatile species) coated with volatile SOA, the AMS may not detect the non-volatile core left after treatment in the TD. The scanning DMA and CCN systems would detect the presence of the non-volatile core, but there would likely not be a corresponding signal in the AMS. How would these results be interpreted? This situation is one where selecting monodisperse aerosol upstream of the TD would greatly facilitate interpretation of the measurement results.

This is an excellent suggestion. Adding size resolution measurements can enhance our results significantly. As with any experimental setup, there are trade-offs. By adding size resolution, we will lose time resolution, but use of both modes of operation is certainly possible in laboratory experiments that can be repeated. Additional challenges may include the characterization of the resulting low aerosol concentrations by the AMS. However, this approach could provide significant advantages in the interpretation of the results for ambient measurements. We will try to explore this idea in future work.

**(5)** Most volatility studies assume that material volatile at or below the thermodenuder temperature set-point is evaporated to the gas phase. However, it seems that chemical reactions stimulated by the high temperature environment may also change the composition and oxidation state of the particles within the TD. Can you comment on the likelihood of this?

There is always a possibility that chemical reactions can occur in the TD and this is one of the main weaknesses of this approach. In this study, however, the likelihood of chemical reactions occurring is relatively small because the temperature in the TD never rises above 125°C. This is a relatively low value compared to most TD studies. A second test is based on the relatively small changes of the particulate phase composition (based on AMS measurements) as the temperature is increasing. However, none of these tests is conclusive. A detailed characterization of the composition of the remaining organic phase is required to quantify the contribution of reactions to the observed changes. This important point has been added to the Conclusions of the paper.

**(6)** Another general question worth asking is: what is the relevance of volatility studies to the atmospheric aerosol since they are never exposed to most of the temperatures used in TD studies?

The major advantage of the volatility studies based on TDs is that they provide insights about the less volatile organic components. These are usually in the particulate phase at room temperature, but they may evaporate partially at higher atmospheric temperatures or when diluted significantly. The least volatile secondary components, when created by gas-to-particle conversion processes are important for the formation and growth of new particles in the atmosphere. Finally, the evolution of the volatility distribution of the organic aerosol can provide an indirect way to gain insights about the very difficult to measure chemical composition and evolution of these thousands of components. This discussion has been added to the introduction of the paper.

**(7)** Finally, it is not very clear to me why volatility would necessarily be well connected to hygroscopicity.

This is an excellent point. It has been assumed that the bridge connecting these two properties is the oxidation level. It has also been proposed that the link may be so strong that knowledge of the oxidation level may be sufficient to determine both properties. Our work here, among others, suggests that this is not so straightforward and these linkages are quite complex and may be even non-monotonic.

More detailed comments:

**(8)** Page 2, line 10: suggest adding that the gases can produce new particles AND condense on pre-existing particles.

Done.

**(9)** Page 2, line 29: please clarify what 'this SOA system' refers to.

It refers to the α-pinene ozonolysis SOA system. We have changed this in the manuscript.

**(10)** Page 3, line 34: is 'MFR' defined somewhere?

We have added a definition for MFR the first time it appears.

**(11)** Page 4, line 4: suggest ". . .that the hygroscopicity of organic aerosol generally increases with . . ."

Done.

**(12)** Page 5, line 5: were flow mixers employed upstream of the two sample flow splits to ensure the AMS/DMA and CPC/CCN received the same aerosol populations?

We did not use flow mixers. The streams were split using a normal T union. We have clarified this in the manuscript.

**(13)** Page 6, line 20: please explain how 'mass fraction remaining' in the TD is determined using SMPS measurements. How is the density measured to convert the mobility volume distribution to a mass distribution?

To calculate the MFR, we divided the TD mass at each SMPS measurement by an interpolated BP mass using the BP measurements before and after the TD measurements. We used a constant density of 1.4 g cm$^{-3}$ for α-pinene ozonolysis SOA according to the recommendation of Kostenidou et al. (2007). We have updated the manuscript to include this information.

**(14)** Page 7, line 4: it is worth mentioning here the shorter residence time you expect at your maximum TD operating temperature.

We have added this point.

**(15)** Page 7, line 4: please comment on the expected range in residence time within the TD. For the low flow rate and large tube diameter, do you expect laminar flow conditions? If so, wouldn't particles traveling near the centerline experience a residence time roughly half those near the walls? If this is the case, how would this impact your results?

We do expect laminar conditions with a Reynolds number around 10, which would mean that particles traveling near the walls have longer residence times than those near the centerline. The effect of the simplifying assumption of an average residence time has been considered by Cappa (2010). The change in MFR when the more detailed fluid dynamics model was used was a few percent. So its effect on the results of the present study in which the volatility is characterized using logarithmically spaced bins is small.

**(16)** Page 9, line 1: please explain what C* means.

$C*$ is the effective saturation concentration used in the 1D-VBS. We have added an explanation and the corresponding reference in the manuscript.

**(17)** Page 9, line 15: I am confused by the statement that using the 1D-VBS allows for comparison of different TD studies regardless of TD operating conditions. If the physical residence time in a TD is too short to allow complete volatilization, or if the temperature time history within a TD is not represented by the temperature set-point of the study, how would a model allow successful intercomparisons of different TD study results?

As stated in the introduction, thermograms are heavily influenced by several factors (OA concentration, particle size, residence time, etc.) and TD studies use different experimental conditions. This can result in quite different thermograms for the same aerosol even when the same TD is used. So use of a model simulating the aerosol evaporation inside the TD can help "translate" the measurements to the same basis, the aerosol volatility distribution. The 1D-VBS is a good framework for this analysis, but of course other descriptions of the aerosol volatility distributions are possible. We have rewritten this section to make it clearer as to why we recommend using a model to generate volatility distributions to compare between studies.

**(18)** Page 10, line 2: please add the supersaturation value set in the CCN instrument after activation diameter.

Done.

**(19)** Page 10, line 6: 155 +/- 1 nm? This level of uncertainty or standard deviation in the measured activation diameter is a little hard to believe, you are reporting a size variation of only +/-0.06%. If you were to scan monodisperse 155 nm diameter PSL calibration particles 50 times and calculate the variation in the measured peak size, my guess is you would see higher intrinsic measurement uncertainty than 0.06%.

This is a valid point and it is probably a coincidence for these data points. The standard deviation of the various activation measurements was a few nanometers. To avoid misleading the readers about the overall variability of our measurements, we do not show in the main text the variability in this specific case, which is a lot less than the average. The variability of all the measurements is shown in the corresponding figures.

**(20)** Page 10, line 10: or that chemical changes to the particles that occurred within the TD rendered the particles less CCN active. . .?

We have added this to the manuscript.

**(21)** Page 10, line 15: You make a very good point here that trying to generalize relationships between volatility, hygroscopicity/cloud activity and oxidation level is over simplifying the situation. However, it would be extremely useful if such relationships could be developed, perhaps by dividing organic species into certain families or structures with key similarities. Any comments?

This could be very helpful in determining useful relationships for families of organic compounds. There are many different ways to divide the species, but one that is widely used is to group them using PMF. It has been shown that these groups (M-OOA, L-OOA, HOA, etc.) exhibit different properties and we could group them this way. For this specific work, since the OA is from one source, this method is not that useful. We hope to be able to develop such relationships applying the approach suggested here in more complex systems including of course ambient OA.

**(22)** Page 11, line 10: can you comment briefly on how you would extend this analytical approach to unknown composition ambient aerosol? E.g. mixtures of insoluble material, soluble inorganics and soluble SOA?

Our intention is to develop the method enough, so that it can be applied to ambient aerosol. However, we believe that the availability of real-time automatic instrumentation for the measurement of the size-composition distribution of the sub-micrometer ambient aerosol will allow us to work with known chemical composition (with the exception of the organic compounds). The analysis could use the known hygroscopic properties of the inorganic aerosol components and focus on the properties of the organics. There are a number of challenges including the determination of the size distribution of the refractory aerosol components that are not measured by the AMS (though the SP-AMS or the SP2 can help with black carbon), issues related to the mixing state of ambient particles, etc. Techniques like the Positive Matrix Factorization (PMF) could help to split the OA into a few components and then determine their corresponding properties. We have included a few lines in the manuscript on how the method would be applied to ambient aerosol.

**(23)** Page 11, end of page: missing period.

Added.

**(24)** Page 13: in performing the sensitivity analysis – did you systematically eliminate each one of the 18 equations and run 17 studies? Or did you just run one study after picking a single equation to eliminate? How would your sensitivity results change if you 'randomly' added + or – one standard deviation to the experimental data used to constrain the equations? What if you added one standard deviation to all of your data?

We have rewritten the beginning of the sensitivity analysis section to provide more clarity on how we performed this test. We systematically removed each one of the 18 equations and solved the resulting system 18 times. A more detailed uncertainty analysis would involve selections of values from the corresponding distributions of the various parameters, repeating the calculation, and estimating the distribution of the values of the resulting parameters. Perturbing one parameter at a time is a cheap way to do that. Selecting extreme values for all parameters may lead to an overestimation of the uncertainty (this combination is quite unlikely). We believe that our sensitivity analysis approach is adequate for its purpose, which is to identify if one of the measurements is dominating the answer of the problem.

**(25)** Page 13, line 17: suggests

Corrected.

**(26)** Page 14, line 2: if the method has difficulty with low concentrations, does this impact its usefulness for studies of the ambient aerosol?

The "low" is used in a relative not an absolute sense here. We now clarify in the revised paper that the mass fraction of a volatility bin must be greater than 0.1 to allow the accurate estimation of the bin's properties based on a single experiment. When multiple experiments are combined together, the method is able to accurately estimate the bins' properties as long as there was enough material (more than 10%) from that bin in at least one experiment.

**(27)** Page 14, it would be helpful if you rewrote the paragraph starting at line 6 so it was easier to understand.

We have rewritten this paragraph to help make it clearer to the reader.

[revised manuscript text omitted]

Several studies have attempted to relate two of the three properties, but few have attempted to relate all three. Jimenez et al. (2009) proposed that the hygroscopicity of OA generally increases with the oxidation level expressed by the O:C ratio and that there is also an inverse relationship between the O:C ratio and volatility. Tritscher et al. (2011) used a volatility and hygroscopicity tandem differential mobility analyzer (V/H-TDMA) and an Aerodyne High Resolution Time-of-Flight Aerosol Mass Spectrometer (HR-ToF-AMS, hereafter AMS) during the chemical aging of $\alpha$-pinene SOA and found that volatility decreased while hygroscopicity and the O:C ratio remained fairly constant. Cerully et al. (2015) used a TD followed by a CCN counter, a scanning mobility particle sizer (SMPS), and an AMS in parallel and observed small changes in hygroscopicity for ambient OA components with dramatically different volatilities and concluded that the more volatile compounds were more hygroscopic than the remaining material. Several other studies (Poulain et al., 2010; Kuwata et al., 2011; Hong et al., 2014; Hildebrandt Ruiz et al., 2015) have investigated the effects of environmental parameters on one or all of these properties. However, these results are often inconclusive or even contradictory and the links among these three properties are yet to be elucidated.

A theoretical framework (Nakao, 2017) has attempted to relate these three properties using the 2D-VBS framework (Donahue et al., 2011). The approach utilized correlations between the O:C ratio, volatility, and hygroscopicity to predict lines of constant $\kappa$ in the 2D-VBS. The study concluded that relatively volatile OA components with a low O:C ratio can have the same hygroscopicity as OA with lower volatility and a higher O:C ratio.

One major obstacle pertaining to an experimentally-determined relationship between these three properties for ambient aerosols is the often unknown or uncertain composition. The

AMS can quantify the concentrations of the non-refractory inorganic aerosol components as well as the total organic aerosol concentration, with further separation of the OA into components provided by techniques like  positive matrix factorization (PMF). However, the analysis of the combined measurements of all three properties remains challenging.

The purpose of this work is the development of a method for the synchronous measurement of OA hygroscopicity, oxidation level, and volatility. In the next section, we describe the  technique that utilizes a suite of aerosol instrumentation to measure these properties. Then, the method is tested with $\alpha$-pinene ozonolysis SOA. This SOA system has been studied extensively so it can be a useful first test for the proposed experimental approach. Our objective is not to perform a comprehensive study of the properties of this SOA (which depend on SOA levels, relative humidity, etc.), but rather to use it as a pilot study. Finally, a data analysis technique is developed to interpret and synthesize the corresponding measurements.

**2 Methodology**

**2.1 Instrument setup**

A schematic of the  experimental setup can be seen in Fig. 1. Particles are sampled through either a TD or a by-pass (BP) line via two three-way valves and then the sample stream is split using a T union between an AMS and a differential mobility analyzer (DMA, TSI, model 3081). The stream from the DMA is split again with a T union between a condensation particle counter (CPC, TSI, model 3010/72) and a CCN counter (CCNC, Droplet Measurement Technologies). In this study, the AMS used a flow rate of 0.1 L min$^{-1}$ while the CPC and the CCNC used 0.3 and 0.5 L min$^{-1}$ respectively. The sheath flow in the DMA was set to 8 L min$^{-1}$ to allow for a 10:1 sheath to aerosol flow ratio as the particles were classified. The upscan of the DMA was set to 120 s and the downscan was set to 15 s.

 a typical experiment, once particles are ready for sampling, they pass through the TD at the first temperature set point for 15-25 min. Then, the particles are sent through the BP as the temperature of the TD increases  (another

15-25 min). Once the TD reaches the next set point, the particles are directed through the TD for the same sampling period  and this process is repeated until measurements at all desired temperatures  have been obtained. Total sampling time for 4–5 temperature set points is around 2.5–3.5 hr. Set points for the TD start at the lowest temperature and always increase.

**2.2 Hygroscopicity**

Hygroscopicity measurements were made with a CCNC, which generates supersaturations by exploiting water's higher mass diffusivity than heat's thermal diffusivity in air (Roberts and Nenes, 2005). The CCNC's fast response time allows it to be coupled to an SMPS, a technique called Scanning Mobility CCN Analysis (SMCA, Moore et al., 2010). In our experiments, polydisperse aerosol was charged with a Po-210 neutralizer and then entered a DMA where the particles were classified by their electrical mobility and counted by a CPC and the CCNC as the DMA voltage was scanned. Particle number concentrations (CN) and size distributions were obtained from the SMPS using the AIM software. CCN concentrations and size distributions were obtained following the technique described in Moore et al. (2010). The SMPS and CCNC size distributions were aligned by matching the minimum in concentration that occurs during the transition between the DMA upscan and downscan. An activation curve was produced by  dividing the CCN concentration by the  (CN concentration) at each size. The activation diameter was calculated by fitting the activation curve to a sigmoidal function:

$$\tag{1}$$

[revised manuscript text omitted]